# Water-Soluble Polymer Polyethylene Glycol: Effect on the Bioluminescent Reaction of the Marine Coelenterate Obelia and Coelenteramide-Containing Fluorescent Protein

**DOI:** 10.3390/ijms24076345

**Published:** 2023-03-28

**Authors:** Tatiana S. Siniakova, Alexander V. Raikov, Nadezhda S. Kudryasheva

**Affiliations:** 1Biophysics Department, Siberian Federal University, 660041 Krasnoyarsk, Russia; 2Institute of Biophysics SB RAS, FRC KSC SB RAS, 660036 Krasnoyarsk, Russia

**Keywords:** polymer, polyethylene glycol, bioluminescence, marine coelenterate, photoluminescence, fluorescent protein, bioassay, toxicity

## Abstract

The current paper considers the effects of a water-soluble polymer (polyethylene glycol (PEG)) on the bioluminescent reaction of the photoprotein obelin from the marine coelenterate *Obelia longissima* and the product of this bioluminescent reaction: a coelenteramide-containing fluorescent protein (CCFP). We varied PEG concentrations (0–1.44 mg/mL) and molecular weights (1000, 8000, and 35,000 a.u.). The presence of PEG significantly increased the bioluminescent intensity of obelin but decreased the photoluminescence intensity of CCFP; the effects did not depend on the PEG concentration or the molecular weight. The photoluminescence spectra of CCFP did not change, while the bioluminescence spectra changed in the course of the bioluminescent reaction. The changes can be explained by different rigidity of the media in the polymer solutions affecting the stability of the photoprotein complex and the efficiency of the proton transfer in the bioluminescent reaction. The results predict and explain the change in the luminescence intensity and color of the marine coelenterates in the presence of water-soluble polymers. The CCFP appeared to be a proper tool for the toxicity monitoring of water-soluble polymers (e.g., PEGs).

## 1. Introduction

In recent decades, pollution of the world’s oceans by water-soluble and insoluble polymers has become a challenge for ecologists. It is known that water-soluble polymers can change the rates of cellular processes via stabilizing biological structures. It is shown [1,2,3] that enzymes included in the gelatin or starch gel matrix become more stable and lead to an increase in the activity. Moreover, water-soluble polymers such as polyethylene glycol (PEG) may cause toxic effects on living organisms [4,5]. The toxic effects of polymer pollution are now being intensively studied [6,7,8,9,10].

Bioluminescence assay systems are widely used in ecological science to monitor the toxic effect of chemicals on living organisms. Bioluminescence intensity is considered as a test physiological parameter for evaluating toxic effects. Simplicity, sensitivity, and a high registration rate of the emission intensity make bioluminescent tests convenient and widely applied. The marine luminescent bacterium is one of the most common bioassays that has been widely used for more than 50 years to monitor the toxicity of media due to its high sensitivity to toxicants [11,12,13]. Another type of bioluminescent assay, the bacterial bioluminescent enzyme system, was suggested in 1990 [14]. An advantage of the enzymatic assays is the possibility to change the sensitivity to toxic compounds by (a) varying the component concentrations and (b) constructing polyenzymatic coupled systems [15,16,17]. Technological applications of the bioluminescent enzymatic system were reviewed in [18,19].

A question arises whether it is possible to detect suppressive or activation effects of chemicals by a simpler system than a bacterial one. Recently, the authors have suggested an application of luminescent proteins to monitor the toxicity in biological liquids [20]. The paper reviews the effects of radiation [21,22], chemical agents [23,24], and higher temperature destruction [25] on the photoluminescence spectra of a coelenteramide-containing fluorescent protein (CCFP). It was stated that CCFPs could serve as new fluorescence biomarkers with color differentiation to explore the results of destructive exposures. CCFP demonstrated changes in color under the exposures listed before. The question of interest is: will the photoprotein obelin and CCFP—a product of the bioluminescent reaction of obelin—be sensitive to water-soluble polymers, for example, PEGs? The color and the intensity of the luminescence are of interest in both cases.

Note that the studies [20,21,22,23,24,25] applied CCFP, a product of the bioluminescent reaction of photoprotein obelin from the marine coelenterate *Obelia longissima*. The structure and mechanisms of the photoprotein bioluminescence are now being intensively studied [26,27,28]. Our previous studies did not find changes in the bioluminescence spectra of the photoprotein obelin after the exposures listed before (chemical, radiation, or thermal); only the bioluminescence intensity appeared to be sensitive to the exposures. The sensitivity of the obelin bioluminescence to PEG is a problem of special interest. 

In general, CCFPs can be isolated from luminous marine coelenterates, e.g., jellyfish *Aequorea* [28] and *Phialidium* (*Clytia*) [29], hydroid *Obelia longissima* [30], etc. A fluorophore of CCFPs is coelenteramide, an external molecule; being a heteroaromatic fluorescent compound, it is non-covalently bound to a protein inside its hydrophobic cavity. The chemical structure of the coelenteramide molecule (neutral and ionized forms) is presented in Figure 1. Coelenteramide is a photochemically active molecule, as it is able to be a proton donor in its electron-excited states and to generate several forms which differ in energy fluorescent states [31] and, hence, in fluorescence color. Contributions of the forms to visible fluorescence spectra depend on the efficiency of the photochemical process and these are governed by the microenvironment of coelenteramide [32,33,34,35,36,37]. There occur similar proton transfer processes and formations of fluorescent forms after chemical/biochemical excitation in the course of bioluminescence reactions in marine coelenterates. 

According to [39], the spectra of the obelin bioluminescence and light-induced fluorescence of CCFP are a superposition of spectral components (emitters) that correspond to different forms of coelenteramide. The contributions of the spectral components might change, indicative of proton interactions in the active center of the protein complex. The contribution of the spectral components to the integral spectrum determines the color of the luminescence. 

The spectra of the obelin bioluminescence and the light-induced fluorescence of CCFP can be deconvolved into Gauss components [39]. It is shown in these studies that the spectra can include three components in the visible region, with the maxima (Figure 2) corresponding to violet, blue–green, and green spectral regions. According to [31,32,33,34,35,36], component I was attributed to the neutral coelenteramide form, while components II and III were attributed to the ionized forms of coelenteramide (Figure 1 and Figure 2). Ionized forms II and III can differ in the effective location of a proton in the complex of polypeptide with coelenteramide between the phenolic coelenteramide group as a proton donor and His22 as a proton acceptor; hence, forms II and III can differ in ionization degree.

There is a possibility to change the spectral characteristics of the photoprotein obelin by varying the rigidity of its environment. The effects of water-soluble polymers on the bioluminescence reaction of obelin and the light-induced fluorescence of its product, CCFP, have not been studied yet. Dependencies of the effects on the polymer characteristics in solutions (molecular weight and concentration) might elucidate the influence of water-soluble polymers on water organisms in the world’s oceans. 

This study aims at evaluating the effects of PEGs of different molecular weights and concentrations on the bioluminescent reaction of the photoprotein obelin from the marine coelenterate *Obelia longissima* and photoluminescence of CCFP, a bioluminescence reaction product. Accordingly, two main points were under consideration: (1) effects of PEGs on the intensity of the bioluminescence and photoluminescence responses; (2) effects of PEGs on the spectra of the obelin bioluminescence and CCFP photoluminescence. The work is original in the area of responses of marine coelenterates to water-soluble polymer pollutants.

## 2. Results and Discussion

We studied the effects of PEGs of different molecular weights (1 kDa, 8 kDa, and 35 kDa) on the bioluminescence of the photoprotein obelin and photoluminescence of the coelenteramide-containing fluorescent protein (CCFP). The concentration of PEG varied from 0 to 1.44 mg/mL. The bioluminescence/photoluminescence intensities/yields and spectra were studied. 

### 2.1. PEG Effect on the Bioluminescence Reaction of Obelin

#### 2.1.1. PEG Effect on the Bioluminescence Yield of Obelin

The time-dependent change of the relative bioluminescence quantum yield is presented in Figure 3. This figure illustrates the results of exposure to three concentrations of PEG; PEG of 1 kDa was chosen as an example.

It can be seen that PEG increases the quantum yield of bioluminescence during all the time periods of registration (relative quantum yield > 1). The bioluminescence intensity increase might be due to the protein complex stabilization in the course of the bioluminescent reaction and to a decrease in the non-radiative relaxation in the electron-excited states of the bioluminescence emitter. We observed an increase in the bioluminescence yields for all the three PEG samples (1 kDa, 8 kDa, and 35 kDa) at all the PEG concentrations applied. 

#### 2.1.2. PEG Effect on the Spectra of Bioluminescence Reaction

The impact of the PEG samples (1 kDa, 8 kDa, 35 kDa) on the bioluminescence spectra was analyzed. Three concentrations of PEGs were analyzed: 0.01 mg/mL, 0.1 mg/mL, and 1 mg/mL. As an example, Figure 4 shows the changes in the bioluminescence spectra in the presence of PEG, 1 mg/mL (8 kDa).

The changes in the spectra upon the addition of PEG are evident from Figure 4, with a decrease in the violet component contribution. We analyzed the effect of PEGs on the contributions of the components to the bioluminescence spectra. The complex bioluminescence spectra in the PEG solutions were deconvoluted into peak components: violet (maximum 400 nm), blue–green (maximum 485 nm), and green (maximum 557 nm). For all the PEG samples, the contributions of the components to the reconstituted bioluminescence spectral outlines were calculated. As an example, Figure 5 shows the kinetics of the violet component contribution in the solutions of PEG, 8 kDa. Three concentrations of the PEG solutions are presented. Similar results were obtained with polymers of the other molecular weight, 1 and 35 kDa.

Figure 5 demonstrates the kinetics of the violet component contribution during the bioluminescent reaction. One can see the suppression of the violet component at the beginning and at the end of the bioluminescent reaction and its increase at the 5th–10th second, which is as high as 190%. Contribution of the sum of II (blue–green) and III (green) components varied as well but were opposite to the I (violet) component. A similar tendency was observed in the solutions of different PEG concentrations. No monotonic concentration dependence was observed, as seen in Figure 5. 

Bioluminescence kinetics in Figure 5 suggests that PEG initiates the less effective deprotonation of coelenteramide in the course of the reaction compared with the initial and final stages of the reaction. It is evident that the binding of photoprotein with fragments of PEG can stabilize the photoprotein structure with a lower efficiency of proton transfer; this process is time-dependent on the bioluminescence kinetics. Details of this dependence should be further studied in special experiments.

Figure 6 presents the concentration dependences of the violet contribution for the different concentrations and molecular weight of PEG. The bioluminescence spectra were analyzed at the beginning (0.6 s), in the middle (6 s), and at the end (12.6 s) of the reaction. The figure demonstrates the absence of the violet contribution dependence on the PEG molecular weight and confirms its absence at the PEG concentration chosen in the experiments. 

The increase in the violet contribution is a result of a decrease in the efficiency of proton transfer in the enzyme-bound coelenteramide in the course of the bioluminescence reaction (Figure 1 and Figure 2): “discharging” (i.e., reconstruction) of the photoprotein complex in the presence of Ca^2+^. Collisional interactions with polymeric fragments and the change in the rigidity of the medium in the presence of PEG are likely to be responsible for this effect. The time-dependence of the violet component during the bioluminescence reaction should be investigated further; the reasons for this are to be clarified in detail. 

The results can depend on temperature, as it is supposed that diffusion components in polymer solutions can be valuable in molecular mechanisms of the polymer interaction with photoprotein. The temperature dependences of the spectral component contributions should also be a subject of further investigations. 

### 2.2. PEG Effect on the Photoluminescence Spectra of the Coelenteramide-Containing Fluorescent Protein

#### 2.2.1. PEG Effect on the Photoluminescence Intensity of the Coelenteramide-Containing Fluorescent Protein

The effect of PEG on the photoluminescence intensity of the coelenteramide-containing fluorescent protein (CCFP) was studied. Figure 7 illustrates the interaction results of CCFP and PEG (1, 8, 35 kDa) at different PEG concentrations; the photoluminescence decay was observed, but no dependence on polymer molecular weight was found. These results can be attributed to the collisional interactions of CCFP with the fragments of the polymer chains. 

The differences in the effects of PEG on CCFP and bioluminescence reactions could be concerned with post-reaction stabilization of the product of the reaction, CCFP. This supposition is based on differences in structures (and hence sensitivities to exogenous compounds) of CCFP and bioluminescence emitters [32].

Therefore, the coelenteramide-containing fluorescent protein appeared to be a proper tool for toxicity monitoring of water-soluble polymers, PEGs. Photoluminescence inhibition efficiency reached 40% at a PEG concentration of 0.15 mg/mL. 

Our results make us suppose that it is not a polymeric molecule as a whole that is responsible for the luminescence suppression but only fragments of the polymeric chains.

#### 2.2.2. PEG Effect on the Photoluminescence Spectra of the Coelenteramide-Containing Fluorescent Protein

Figure 8 provides the normalized photoluminescence spectra of the coelenteramide-containing fluorescent protein (CCFP) at different PEG concentrations. The concentrations applied are presented in Figure 7. PEG of 8 kDa was chosen here as an example. 

It might be seen from Figure 8 that no changes in the photoluminescence spectra of CCFP upon the variation of the PEG concentration were observed. Spectral maxima (510 nm) did not shift and the shape of the spectra changed negligibly. This indicates that PEGs do not affect the efficiency of photochemical proton transfer in the CCFP complex, nor do they affect the ratio of the protonated and deprotonated forms of coelenteramide in the fluorescent protein (Figure 1 and Figure 2). Similar results were obtained for polymers of other molecular weights, e.g., 1 and 35 kDa.

Hence, only the bioluminescence reaction spectra appeared to be sensitive to the presence of the polymer rather than the product of the bioluminescence reaction: “discharged photoprotein” or coelenteramide-containing fluorescent protein. This indicates the optimality and stability of the protein structure.

## 3. Materials and Methods

### 3.1. Materials

Recombinant preparations of obelin D12C were obtained from the Photobiology Laboratory, Institute of Biophysics, SB RAS, Krasnoyarsk, Russia. A detailed description of the recombinant obelin production was given in [40,41]. Solutions of lyophilized obelin (2.65 mg/mL) in a 0.02 M tris(hydroxymethyl)aminomethane buffer (pH 7) and 0.05 M ethylenediaminetetraacetic acids (Sigma-Aldrich, St. Louis, MO, USA) were applied. Three PEG samples of different molecular weights were used: 1, 8, and 35 kDa. Table 1 provides the physicochemical characteristics of the polymer samples. 

### 3.2. Instrumentation

Bioluminescence and light-induced fluorescence spectra were measured with a Cary Eclipse-2000 spectrofluorometer (Agilent, Santa Clara, CA, USA). Ultraviolet (UV) radiation was emitted by the xenon flash lamp (80 Hz), producing 80 flashes per second [44]. Emission spectra for bioluminescence and photoluminescence were recorded in the range from 370 to 600 nm; photoluminescence excitation was 350 nm. The scanning rate was 600 nm/min and the spectral bandwidths for emission and excitation were 10 nm and 10 nm, with the wavelength accuracy being ± 1.5 nm. A quartz cuvette with a rectangular cross-section (2 × 10 mm) was used to register the spectra. The emission spectra were converted from the wavelength- to wavenumber-dependences, as presented in [45].

### 3.3. Experiment Methodology 

We examined the PEG effects on the bioluminescent reaction of the marine coelenterate Obelia and coelenteramide-containing fluorescent protein. The amount of 2 µL of PEG of different concentrations and molecular weights was added to the recombinant obelin preparation (10^−6^ mg/mL). The bioluminescence reaction was triggered by 10 µL Ca^2+^. The time of bioluminescence registration varied from 1 to 12.6 s. 

The light-induced fluorescence spectra of Ca^2+^-discharged obelin (i.e., photoluminescence of the coelenteramide-containing protein) were measured. The excitation wavelength was 350 nm, with the ambient temperature being 20–25 °C. 

The changes in bioluminescence and photoluminescence spectra were studied in five experiments, providing statistical significance.

### 3.4. Analysis of the Obelin Bioluminescence Spectra

The complex spectra were deconvoluted into peak components by the peak analysis using Origin lab 2018. The mathematical treatment involved two steps: 

The function increment method [46], based on the Gauss distribution, was applied to identify spectral components.

A secondary derivative for the fluorescence intensity was calculated [47] to determine the number of spectral components and their maxima. 

To compare the squares of the calculated and experimental spectra, the d-values were calculated as follows: *d* = (|*S_exp_* − *S_sum_*|/*S_exp_*) × 100%,(1)
where *S_exp_* is the square of the experimental spectral outline and *S_sum_* is the square of the sum of the reconstituted spectral outlines. The *d* values for all the spectra did not exceed 10%. All the spectra were deconvoluted into a minimum number of components in the coordinates of luminescent intensity and wavelength number. Microsoft Office Excel was used to carry out the statistical analysis.

An example of the deconvoluted spectrum is presented in Figure 2 in the Introduction. 

Relative bioluminescence quantum yields were evaluated as *S_PEG_*/*S_contr_*_,_ where *S_PEG_* and *S_contr_* are the squares of the experimental spectral outlines in the presence and absence of PEG, respectively, in the coordinates of luminescent intensity and wavelength number. 

## 4. Conclusions 

Generally speaking, studies of the effects of polymers on living marine systems are now of vital importance due to the intensive pollution of the world’s oceans with polymeric compounds, including commercial water-soluble polymers. The studies should develop in different directions; marine coelenterates and their protein complexes are significant part of these investigations. The following direct conclusions can be made from the results presented in the current paper:

1. PEGs increase the intensity of the bioluminescence reaction because of the photoprotein structure stabilization. The efficiency of the bioluminescence activation does not depend on PEG molecular weights (1, 8, or 35 kDa) or the PEG content in the interval of PEG concentrations 0.01–1 mg/mL.

2. PEGs initiate spectrum shifts and changes in the contribution of spectral components in the course of the bioluminescent reaction of obelin. The changes are multidirectional and depend on the reaction stage. The increase in the contribution of the blue component is as high as 190%.

3. PEGs decrease the photoluminescence intensity of the coelenteramide-containing fluorescent protein, which is likely due to collisional interactions with the fragments of the polymer chains.

4. PEGs do not change the shape of the photoluminescence spectrum of the coelenteramide-containing fluorescent protein. This may indicate that PEG does not affect the efficiency of proton phototransfer in the protein complex.

5. The coelenteramide-containing fluorescent protein appeared to be a proper tool for the toxicity monitoring of water-soluble polymers, PEGs, which include polar and nonpolar chemical groups and hence are able to interact with different fragments of biological structures, producing integral bioresponses.

## Figures and Tables

**Figure 1 ijms-24-06345-f001:**
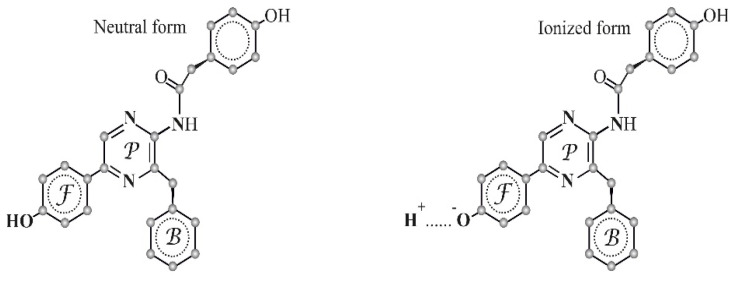
The chemical structure of the coelenteramide molecule (neutral and ionized forms). The aromatic fragments that can be involved in electronic excitation are marked with the letters F, B, and P, corresponding to phenolic, benzene, and pyrazine rings [38].

**Figure 2 ijms-24-06345-f002:**
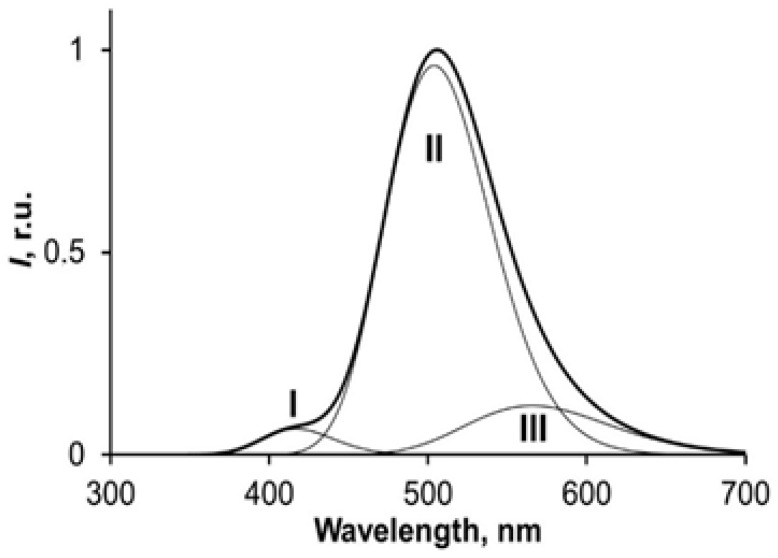
Example of the deconvolution of the spectra of coelenteramide-containing fluorescent protein into Gauss components (I–III); 350 nm photoexcitation.

**Figure 3 ijms-24-06345-f003:**
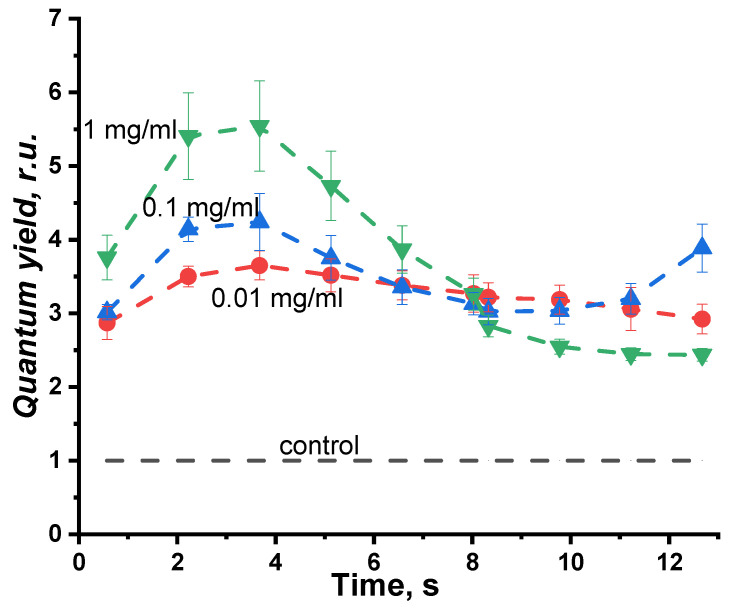
Change in the relative bioluminescence quantum yield vs. time of the bioluminescence reaction. Polyethylene glycol 1 kD, concentrations: 0.01, 0.1, and 1 mg/mL.

**Figure 4 ijms-24-06345-f004:**
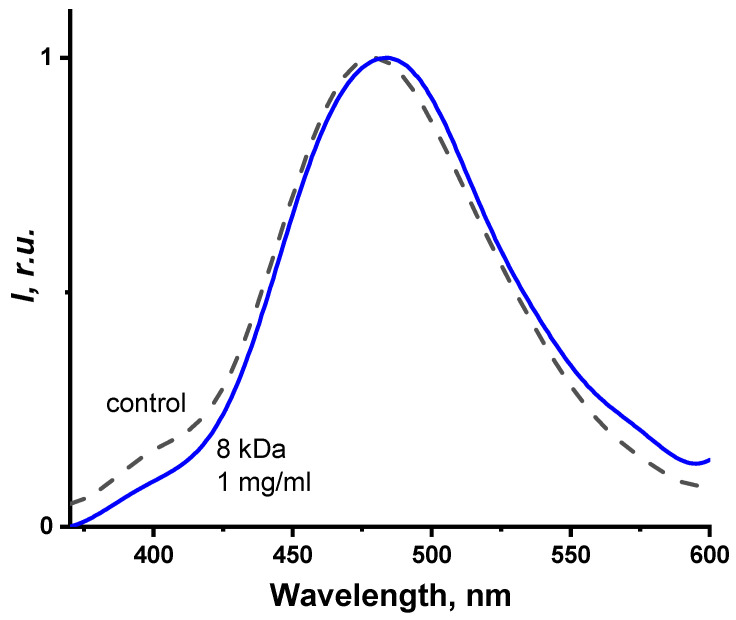
Bioluminescence spectra of obelin exposed to 1 mg/mL of polyethylene glycol, 8 kDa. The time of registration was 0.6 s.

**Figure 5 ijms-24-06345-f005:**
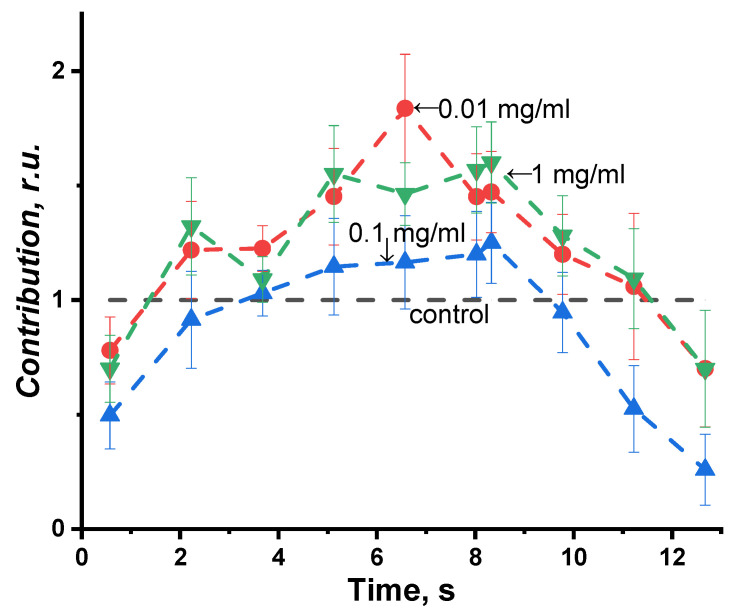
Kinetics of the violet component contribute to the bioluminescence spectra in the solutions of polyethylene glycol, 8 kDa. The PEG concentrations: 0.01, 0.1, and 1 mg/mL.

**Figure 6 ijms-24-06345-f006:**
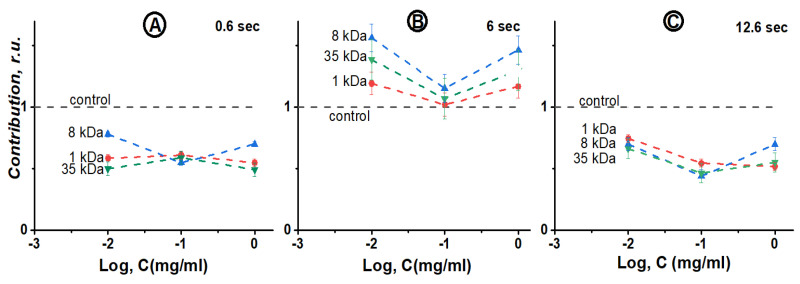
Dependence of the violet component contribution on polyethylene glycol (PEG) concentration at the registration time 0.6 s (**A**), 6 s (**B**), 12.6 s (**C**). The PEG molecular weights: 1 kDa (red), 8 kDa (blue), and 35 kDa (green), with the registration wavelength of 485 nm.

**Figure 7 ijms-24-06345-f007:**
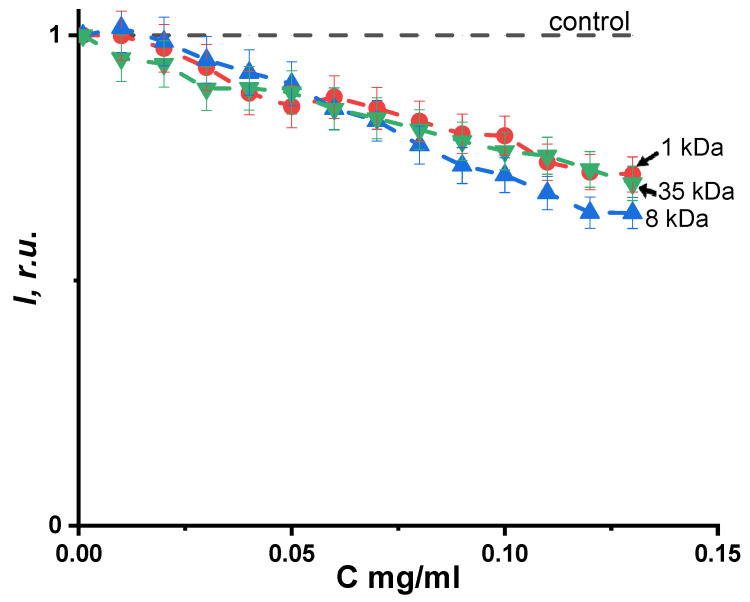
Dependence of the photoluminescence intensity, *I*, of the coelenteramide-containing fluorescent protein on the polyethylene glycol concentration, with an excitation wavelength of 350 nm and a registration wavelength of 510 nm.

**Figure 8 ijms-24-06345-f008:**
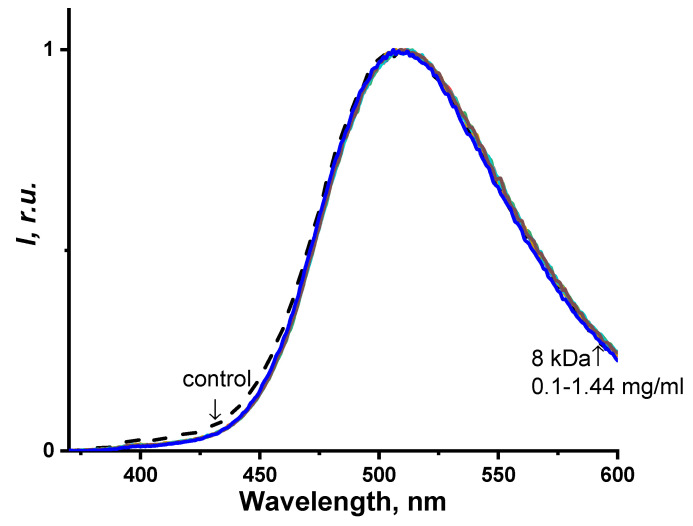
Normalized spectra of the photoluminescence intensity of the coelenteramide-containing fluorescent protein with different concentrations of polyethylene glycol, 8 kDa. The concentrations applied are presented in Figure 6. The excitation wavelength was 350 nm.

**Table 1 ijms-24-06345-t001:** Physicochemical characteristics of PEG of different molecular weights.

	PEG 1000	PEG 8000	PEG 35000
Formula	(C_2_H_4_O)_n_H_2_O [42]		
pH 1% solution	5.0–7.0		
Molecular weight	900–1100	7200–8200	35,000–40,000
Solubility	Soluble in organic solvents. Solubility decreases with increasing the molecular weight of the polymer [43].		
Reactivity	Mostly inert, can form complex compounds with salts of alkaline/earth metals [43].		
Toxicity	Non-toxic [4].		

## Data Availability

Not applicable.

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
