# Peer review of "Water-Soluble Polymer Polyethylene Glycol: Effect on the Bioluminescent Reaction of the Marine Coelenterate Obelia and Coelenteramide-Containing Fluorescent Protein"

_ijms, 2023, doi:10.3390/ijms24076345_

Round 1
Reviewer 1 Report
This work proposed an interesting method that analyzing bioluminescence spectra to monitor water-soluble polymers, PEGs. This paper may be accepted after minor revisions solving questions as follow.
1. Authors used specific peak intensity to illustrate change of component. But luminescence spectra are complicated in real world, mixed with different signals from different components. Can this method keep effective in reality?
2. Do the peaks intensity vary when temperature change and time change? As authors pointed out that quantum yield changes largely when time changes, reviewer do not think this method has high precision.
3. Fig. 8, it seems that peak positions of spectra of different concentration do not vary. But we can see that intensity around 425 nm is smaller than control, so it may be inappropriate to say that ‘PEGs do not affect the ratio efficiency of photochemical proton transfer in the CCFP complex’. More illustration is needed about the phenomenon around 425nm.
Author Response
We are highly grateful to the reviewer for favorable comments and suggestions to improve our manuscript. We’ve changed some parts of the manuscript and marked it with color. Remarks of Reviewer and our responses follow in attached document.

Reviewer 2 Report
This article is devoted to the study of the effect of polyethylene glycol on the bioluminescent reaction of the marine coelenteria Obelia and celenteramide-containing fluorescent protein. The article is written in a clear and accessible language. the main ideas and results are beyond doubt. By topic, this article fits under the section "Molecular Biophysics". There are some points that you should pay attention to:
1. Please unify all drawings.
2. Some parts have text highlighting. It is not necessary.
3. The data obtained by the authors should be compared in more detail with those known from the literature. This will slightly expand the description and make it more fundamental.
4. From the text of the article it is not entirely clear why the authors take a polymer with a mass of exactly 8 kDa for some analyses.
5. Figure 7. Why are similar effects observed for 1 kD and 35 kD polymers? What is the reason for the non-linearity?
6. Conclusions can be shortened and made more concise.
Author Response

(The authors gave the same response as above.)

Reviewer 3 Report
In this work, Siniakova and co-workers studied the effect of the water-soluble polymer polyethylene glycol on both the bioluminescent reaction of the marine coelenterate Obelia and on coelenteramide-containing fluorescent protein. They have found that PEG had an opposite behavior on both systems, by increasing the intensity of the former and by decreasing that of the latter. PEG was also found to change the spectra of the bioluminescent reaction, while having no effect on the spectra of CCFP. The authors have also found that these effects appear to not be dependent on the concentration and molecular weight of PEG.
These results are interesting and provide a novel framework to modify the properties of this relevant bioluminescent system. It can also provide a basis for novel applications regarding the monitoring of such water-soluble polymers. Thus, this work could be of interest for the readership of this journal. However, there are some aspects that need further revision:
-Did the authors evaluate the effect exerted by other water-soluble polymers? This is important to see if some of these results are specific to the addition of PEG;
-The authors stated that "The coelenteramide-containing fluorescent protein appeared to be a proper tool for toxicity monitoring of water-soluble polymers, PEGs". The authors should explain a bit better why they think so.
-The two ionization states of Clmd present in Figure 1 are not the only ones possible;
-The energy diagram in Fig. 1 is quite simplistic and not particularly informative. It should either be improved or removed;
-To what coelenteramide forms correspond components II and III of Figure 2? Here, the authors talk about 3 forms, while Fig. 1 presents only two;
-Beginning of section 2, the authors stated that they varied PEG concentrations from 0 to 1.44 mg/ml. However, in the various Figures, the highest concentration presented is 1 mg/mL. Why so?
-Fig. 3: it appears to be another increase of yield after 12 s for 0.1 mg/mL. The authors should clarify this.
-In Fig. 2 it is indicated that the spectrum include 3 components. However, for Fig. 4, the authors stated that the spectra were deconvoluted into 2 components. Why this difference?
-Fig. 5 includes the kinetics of violet component contributions to the bioluminescence spectra in the solutions of PEG. Do the contributions of the different components also vary as a function of reaction time for the control?
-It is not so clear what this time-dependence of the violet component really means for the reaction mechanism. Are the authors suggesting processes of initial protonation up to six seconds, followed by deprotonation?
-The authors should also characterize the photoluminescence of just Coelenteramide, with addition of PEG;
-It is not clear why PEG has so different effects on CCFP and the bioluminescent reaction. This should be better explained/discussed;
-The authors state that there is an evident red-shift in the spectra of Fig. 4 with addition of PEG. Is this variation statistically significant, if we take into account replicate measurements?
Author Response
We forward you our reply to Reviewer 3 for our manuscript untitled “Water-soluble polymer polyethylene glycol: effect on bioluminescent reaction of marine coelenterate Obelia and coelenteramide-containing fluorescent protein” to be published in Special Issue "Bioluminescent and Fluorescent Proteins: Molecular Mechanisms and Modern Applications 3.0", Journal of Molecular Sciences.
We are highly grateful to the reviewer for favorable comments and suggestions to improve our manuscript. We’ve changed some parts of the manuscript and marked it with green color.

Round 2
Reviewer 3 Report
While the authors have tried to address my comments, there are still issues that need to be clarified:
-Regarding my previous comment 3, the Figure still does not include the possible ionization states of Clmd. The authors should complete Figure 1 with these states, and then define in the text which ones could be considered for the 3 components of the spectrum;
-The information discussed in my previous comment nine should be made explicit in the manuscript;
-Regarding my previous comment 10, it is still not clear why and how there is an increase in the violet component in the first six seconds, followed by decrease in this contribution. The authors must provide a more clear and detailed explanation/hypothesis;
-Regarding my previous comment 11, how about the fluorescence quantum yield of Clmd? Was it affected?
-Regarding my previous comment 12, in what way post-reaction stabilization lead to different effects?
-Regarding my previous comment 13, it is still not clear if these differences/changes are statistically significative or not.
Author Response
We forward you our reply to Reviewer 3 (second round) for our manuscript untitled “Water-soluble polymer polyethylene glycol: effect on bioluminescent reaction of marine coelenterate Obelia and coelenteramide-containing fluorescent protein” to be published in Special Issue "Bioluminescent and Fluorescent Proteins: Molecular Mechanisms and Modern Applications 3.0", Journal of Molecular Sciences.
We are highly grateful to Reviewer for favorable comments and suggestions to improve our manuscript. We’ve changed some parts of the manuscript and marked it with yellow color. Our previous changes are marked with green. Remarks of Reviewer and our responses follow below.

Round 3
Reviewer 3 Report
The manuscript can be accepted.